# The Role of Hedgehog Signaling in the Melanoma Tumor Bone Microenvironment

**DOI:** 10.3390/ijms24108862

**Published:** 2023-05-16

**Authors:** Karnoon Shamsoon, Daichi Hiraki, Koki Yoshida, Kiyofumi Takabatake, Hiroaki Takebe, Kenji Yokozeki, Naohiro Horie, Naomasa Fujita, Nisrina Ekayani Nasrun, Tatsuo Okui, Hitoshi Nagatsuka, Yoshihiro Abiko, Akihiro Hosoya, Takashi Saito, Tsuyoshi Shimo

**Affiliations:** 1Division of Reconstructive Surgery for Oral and Maxillofacial Region, Department of Human Biology and Pathophysiology, School of Dentistry, Health Sciences University of Hokkaido, Tobetsu 061-0293, Japan; 2Division of Clinical Cariology and Endodontology, Department of Oral Rehabilitation, School of Dentistry, Health Sciences University of Hokkaido, Tobetsu 061-0293, Japan; 3Division of Oral Medicine and Pathology, Department of Human Biology and Pathophysiology, School of Dentistry, Health Sciences University of Hokkaido, Tobetsu 061-0293, Japan; 4Department of Oral Pathology and Medicine, Graduate School of Medicine, Dentistry and Pharmaceutical Sciences, Okayama University, Okayama 700-8525, Japan; 5Division of Histology, Department of Oral Growth and Development, School of Dentistry, Health Sciences University of Hokkaido, Tobetsu 061-0293, Japan; 6Division of Dental Anesthesiology, Department of Human Biology and Pathophysiology, School of Dentistry, Health Sciences University of Hokkaido, Tobetsu 061-0293, Japan; 7Department of Oral and Maxillofacial Surgery, Faculty of Medicine, Shimane University, Izumo 693-8501, Japan

**Keywords:** tumor bone microenvironment, malignant melanoma, Hedgehog, Gli

## Abstract

A crucial regulator in melanoma progression and treatment resistance is tumor microenvironments, and Hedgehog (Hh) signals activated in a tumor bone microenvironment are a potential new therapeutic target. The mechanism of bone destruction by melanomas involving Hh/Gli signaling in such a tumor microenvironment is unknown. Here, we analyzed surgically resected oral malignant melanoma specimens and observed that Sonic Hedgehog, Gli1, and Gli2 were highly expressed in tumor cells, vasculatures, and osteoclasts. We established a tumor bone destruction mouse model by inoculating B16 cells into the bone marrow space of the right tibial metaphysis of 5-week-old female C57BL mice. An intraperitoneal administration of GANT61 (40 mg/kg), a small-molecule inhibitor of Gli1 and Gli2, resulted in significant inhibition of cortical bone destruction, TRAP-positive osteoclasts within the cortical bone, and endomucin-positive tumor vessels. The gene set enrichment analysis suggested that genes involved in apoptosis, angiogenesis, and the PD-L1 expression pathway in cancer were significantly altered by the GANT61 treatment. A flow cytometry analysis revealed that PD-L1 expression was significantly decreased in cells in which late apoptosis was induced by the GANT61 treatment. These results suggest that molecular targeting of Gli1 and Gli2 may release immunosuppression of the tumor bone microenvironment through normalization of abnormal angiogenesis and bone remodeling in advanced melanoma with jaw bone invasion.

## 1. Introduction

Melanoma is one of the most aggressive cancers, and its incidence has increased globally over the past 30 years [1]. Melanomas are melanocyte-lineage tumors that are incurable once they have spread, and despite breakthroughs in the treatment of melanomas, the median survival time of melanoma patients is still only 4–6 months [2,3]. Although melanomas account for only 1% of skin cancers, they account for >80% of skin cancer deaths [4]. Melanoma that develops in the mucosa is rare, comprising <2% of the total melanomas diagnosed [5]. Mucosal melanomas have epidemiological and genetic characteristics that differ from those of skin-derived melanomas [6], and the survival of patients with mucosal melanoma is lower than that of patients with skin melanoma [7].

Advanced cases of oral malignant melanoma often involve the tumor’s invasion into the maxilla and/or mandible, with a significantly low 5-year survival rate at 26%; such bone invasion is a prognostic indicator of a poor clinical outcome [8]. Advanced oral melanomas are not expected to respond to chemotherapy, but current first-line surgical treatment, i.e., broad bone resection to remove the tumor mass, frequently impairs a patient’s quality of life. The main postoperative treatments for oral melanomas are irradiation [9], adjuvant chemotherapy [10,11], and adjuvant immunotherapy. However, it has been indicated that irradiation does not effectively increase patients’ survival or reduce the tumor recurrence rate [10,12], whereas patients who have undergone chemotherapy show prolonged survival [13]. Immune checkpoint drugs have demonstrated longer-lasting effects against mucosal melanomas [14,15], although the response rate is still low. The recurrence rate of mucosal melanoma remains high, and the lack of a sufficiently effective treatment has led to evaluations of several post-resection therapies. It is, thus, crucial to (i) determine the precise details of the origin and development of melanoma cells at the genetic level, (ii) identify the downstream target genes, and (iii) create novel and potent medications to suppress melanoma cells’ growth and metastasis. There is also an urgent need for novel treatments for patients with relapsed or refractory melanoma based on new knowledge of the advanced stages of melanomas.

There are two types of signal activation pathways within the Hedgehog (Hh) signaling pathway: canonical and non-canonical signaling [16]. Canonical Hh /Gli (glioma-associated) signaling is triggered by the binding of Hh ligands to the twelve-pass transmembrane receptor Patched 1 (PTCH1). With this binding, PTCH1 no longer represses the seven-pass transmembrane G protein-coupled receptor Smoothened (SMO), thus allowing the intracellular activation of the zinc finger transcription factor Gli2, which translocates into the cell nucleus and transactivates Gli family zinc finger 1 (Gli1) promoter. The aberrant activation of Hh/Gli signaling that occurs in a variety of cancers leads to the activation of Gli transcription factors, which initiate and promote tumor growth via the continuous transactivation of Hh target genes [17]. There have also been reports of non-canonical Gli activation pathways in cancer, which may take place without the involvement of upstream PTCH/SMO signaling [17]. However, the emergence of acquired resistance, significant side effects, and patient relapse after drug discontinuation pose challenges to the effective therapeutic use of SMO antagonists [18]. Cancers such as melanoma that activate both canonical and non-canonical Hh /Gli signaling pathways may not respond well to SMO interference in terms of preventing Hh signaling [19].

GANT61 is a synthetic compound derived from hexahydropyrimidine, and it is notable for its efficient binding to Gli transcription factors as well as to the GliDNA complex [20]. Several investigations have demonstrated that GANT61 significantly decreases the transcriptional production and gene expression of Gli1, PTCH1, and other Hh pathway target genes, as evidenced by inventoried Gli assays in a range of cancer cell types [21]. The tumor-suppressive effect of GANT61 was shown to induce the apoptosis of melanoma cells [19,22] and was described as effective against the resistance of melanoma cells to vemurafenib, an inhibitor of BRAF [23]. These findings suggest that Gli factors play an important role in melanoma progression.

The mechanisms underlying the bone destruction by melanoma involving Hh signaling have not been established, and no published studies have focused on the potential of the Gli inhibitor GANT61 for treating melanoma-induced bone destruction. In the present study, we examined the effects of the Gli1 and Gli2 dual inhibitor, GANT61, in a melanoma bone destruction mouse model and investigated its tumor suppressive mechanism.

## 2. Results

### 2.1. Immunohistochemical Expressions of SHH, Gli1, and Gli2 in Human Melanoma Samples

Before analyzing the mouse model of tumor bone destruction using skin-derived B16 mouse melanoma cells, a screening analysis of Sonic Hedgehog (SHH) signaling was performed using skin melanoma resection specimens without bone involvement (*n* = 37). To determine whether soft tissue melanoma expresses SHH and its signals Gli1 and Gli2, we performed an immunohistochemical analysis of excised human melanoma specimens and normal skin tissues, and we observed the expressions of SHH, Gli1 and Gli2 not only in melanoma cells but also in tumor vascular endothelial cells in the stroma (Figure 1A). The normal skin tissues showed few of these positive cells, and the number of SHH-, Gli1-, and Gli2-positive cells per mm^2^ was significantly higher in the melanoma tissues than in the normal skin tissues (Figure 1B, *p* < 0.05). Notably, we observed strong significant correlations between SHH and Gli1 (*p* < 0.0001, Figure 1C) and between SHH and Gli2 (*p* < 0.0001, Figure 1C) in the immunohistochemical-staining melanoma samples. Together, these results establish a strong correlation between the activation of SHH and the Gli pathway during melanoma progression. To clarify whether SHH and the Gli1 and Gli2 pathway are involved in oral malignant melanoma-induced bone destruction, we next determined the distribution pattern of SHH and its signaling via immunohistochemistry. Figure 1D provides representative microscopic images of invasive bone destruction observed in a patient with oral malignant melanoma in the maxillary region. SHH, Gli1, and Gli2 are highly expressed in tumor cells that have invaded the bone matrix and the tumor vasculature (Figure 1D, triangular arrowheads). Notably, strong expressions of SHH, Gli1, and Gli2 are observed in osteoclasts appearing at the site of jaw bone resorption (Figure 1D, arrowheads).

### 2.2. The Effects of GANT61 Treatment on the Cancer Bone Destruction Mouse Model

Next, to investigate the role of tumor-secreted SHH in bone-destructive lesions of melanoma, we inoculated B16 cells into the bone marrow cavity of the tibia of mice and examined the mice 16 days later using 3D-CT imaging and histological analysis. Compared to the contralateral tibia without tumor inoculation, the group of mice treated with DMSO after tumor inoculation showed osteolytic bone destruction in the lateral cortical bone (Appendix A). Importantly, treatment with the Gli1 and Gli2 dual inhibitor, GANT61, decreased the development of osteolytic area in the lateral cortical bone of the mice inoculated with B16 cells compared to that in the DMSO-treated group. The histological analysis revealed that in the DMSO-treated group of mice inoculated with B16 cells in the tibia, tumor cells invaded the bone marrow cavity and destroyed cortical bone on the marrow side (Appendix A), unlike the GANT61-treated group (Appendix A). The tibiae on the opposite side, at which no B16 cells were inoculated, are presented for comparison, showing that cortical bone is preserved in both the DMSO- and GANT61-treated groups (Appendix A). The length of the ossification zone in the growth plate tended to be shorter in the GANT61-treated group than in the DMSO-treated group (Appendix A). On the other hand, tumor cells in the DMSO-treated group after the inoculation of B16 cells infiltrated into the ossification zone, suggesting that ossification zone formation was inhibited. In the group treated with GANT61 after the inoculation of B16 cells, tumor cell proliferation was suppressed and the formation of the ossification zone tended to be restored (Appendix A).

Since osteoclastic bone resorption is associated with tumor bone destruction in the tumor bone microenvironment, we next examined the distribution of osteoclasts in tumor cell-engrafted tibiae. At 16 days after the inoculation of B16 cells into the tibial metaphysis, the control group treated with DMSO showed tartrate-resistant acid phosphatase (TRAP)-positive osteoclasts on the bone marrow-lined disrupted cortical bone bordering the tumor cells (Figure 2A). In contrast, in the GANT61-treated group, few TRAP-positive osteoclasts were observed in the cortical bone bordering B16 cells in the bone marrow of the tibial metaphysis (Figure 2B), and the number of TRAP-positive osteoclasts per mm^2^ was significantly suppressed by GANT61 (Figure 2D, *p* < 0.05). Very few TRAP-positive osteoclasts were observed in the medial cortical bone in the bone marrow of the tibial metaphysis on the contralateral side where no B16 cells were inoculated in both the DMSO- and GANT61-treated groups (Figure 2A,B). Figure 2C provides histochemical representative images of TRAP staining of tibial growth plate with and without B16 cell inoculation and treated with DMSO or GANT61. Observations of growth plates without B16 cell inoculation revealed that although the formation of ossification zones was suppressed in the GANT61-treated group compared to the DMSO-treated group, the numbers of TRAP-positive cells did not differ significantly between the groups (Figure 2C,D). However, the growth plate of the DMSO-treated B16 cell-inoculated group showed fewer TRAP-positive cells than the DMSO-treated group without B16 cell inoculation (Figure 2C,D, *p* < 0.05), whereas the number of TRAP-positive cells was recovered in the GANT61-treated B16 cell-inoculated group compared to the DMSO-treated B16 cell-inoculated group (Figure 2C,D, *p* < 0.05).

An immunohistochemical analysis of the effect of GANT61 on tumor angiogenesis was thus conducted, using the endothelial marker, endomucin, in the tumor bone microenvironment. When B16 cells were inoculated into the tibial metaphysis, abundant endomucin-expressing endothelial cells in tumor tissue were observed in the bone marrow (Figure 2E). The number of endomucin-expressing endothelial cells in tumor tissue was decreased in the GANT61-treated group (Figure 2E,F), and the number of endomucin-expressing endothelial cells in tumor tissue per unit area was significantly decreased in the GANT61-treated group compared to the control group (Figure 2H, *p* < 0.05). Endomucin staining of contralateral tibial metaphyses without B16 cells was performed for comparison, and no endomucin-positive cells were detected in normal bone marrow cells (Figure 2F). Figure 2G provides immunohistochemical representative images of endomucin staining of tibial growth plates with or without B16 cell inoculation and treated with DMSO or GANT61. Our observations of the growth plates of the group without B16 cell inoculation showed that the number of blood vessels entering the ossification zone from the bone marrow side was suppressed in the GANT61-treated group compared to the DMSO-treated group (Figure 2G). The growth plates of the DMSO-treated group transplanted with B16 cells showed that a large number of endomucin-positive tumor blood vessels invaded into the ossification zone, and that physiological blood vessels invading the ossification zone had disappeared (Figure 2G). The GANT61-treated group showed a decreased number of tumor blood vessels and a recovery of the number of physiological blood vessels entering the ossification zone from the bone marrow side (Figure 2G). However, both tumor blood vessels and physiological blood vessels became endomucin-positive, and distinguishing between them is a challenge for future studies.

To further clarify the effect of GANT61 on tumor cells at the site of bone destruction, we performed an immunohistochemical analysis by staining with anti-Melan-A antibody, a marker for melanoma. In the DMSO-treated group, Melan-A-positive tumor cells filled the bone marrow cavity and spread outward from the cortical bone destruction site (Figure 3A). In contrast, Melan-A-positive tumor cells in the GANT61-treated group were confined to a portion of the bone marrow cavity and were not in contact with the lateral cortical bone (Figure 3A). The number of Melan-A-positive tumor cells per mm^2^ in the GANT61-treated group was significantly suppressed compared to that in the DMSO-treated control group (Figure 3A, *p* < 0.05). No Melan-A-positive cells were observed in the contralateral tibial metaphyses where no B16 cells were inoculated (Figure 3A), indicating that Melan-A specifically recognizes B16 cells.

Proliferation markers are used to determine the behavior and prognoses of malignant tumors, and, in the present study, we assessed the effect of GANT61 on the proliferative potential of B16 cells in the tumor bone microenvironment by determining the percentage of tumor cells stained with proliferating cell nuclear antigen (PCNA) in the paraffin sections. PCNA expression in the nuclei of tumor cells showed strong staining in the tumor cells of the DMSO-treated control group (Figure 3B), whereas PCNA expression in the tumor cells of the GANT61-treated group was more sparse (Figure 3B). The labeling index (LI) in the GANT61-treated group (37.85 ± 1.68%) was significantly suppressed compared to that in the control group (68.8 ± 3.43%) (Figure 3B, *p* < 0.05). PCNA staining in the contralateral tibial metaphyses without B16 cells was performed for comparison and showed no PCNA-positive cells in the bone marrow cells (Figure 3B).

The results so far suggest that GANT61 suppresses tumor cell numbers in the bone microenvironment; this led us to investigate if it is through the inhibition of canonical Hh signaling. SHH-expressing cells in tumor tissue present in the bone marrow were observed when B16 cells had been inoculated into the tibial metaphysis (Figure 3C). The percentage of SHH-positive cells in Melan-A-positive tumor cells was significantly decreased in the GANT61-treated group (LI: 45 ± 5.08%) compared to the control group (LI: 65 ± 2.63%) (Figure 3C, *p* < 0.05). SHH staining of the contralateral tibial metaphysis without B16 cells was performed for comparison, and very few SHH-positive cells were detected in the bone marrow cells (Figure 3C).

Regarding the expressions of Gli1 and Gli2, we observed Gli1 and Gli2 expressions in the tumor tissue in the bone marrow in addition to SHH expression. In contrast, the percentage of Gli1- and Gli2-positive cells in Melan-A-positive tumor cells was significantly decreased in the GANT61-treated group compared to the control group (Gli1: Figure 3D, *p* < 0.05, Gli2: Figure 3E, *p* < 0.05). Similarly, Gli1 and Gli2 staining of the contralateral tibial metaphyses without B16 cell inoculation for comparison showed hardly any Gli1- or Gli2-positive cells in the bone marrow cells (Figure 3D,E). An immunohistochemical analysis of the effect of GANT61 on PD-L1 expression in the tumor bone microenvironment was thus performed, and it revealed that when B16 cells were inoculated into the tibial metaphysis of the mice, PD-L1-expressing cells in tumor tissue present in the bone marrow were recognized (Figure 3F). The percentage of PD-L1-positive cells in Melan-A-positive tumor cells was significantly decreased in the GANT61-treated group as indicated by the LI (34 ± 2.23%) compared to the control group’s LI (72 ± 2.42%) (Figure 3F, *p* < 0.05). PD-L1 staining of the contralateral tibial metaphyses without B16 cells was performed for comparison, and very few PD-L1 positive cells were detected in the bone marrow cells (Figure 3F). These results suggest that GANT61 is involved in the reduction in SHH, Gli1, Gli2, and PD-L1 expressions in surviving melanoma cells in the bone microenvironment.

### 2.3. The Effect of GANT61 on the Proliferation of B16 Cells

To investigate whether GANT61 would have an inhibitory effect on the growth of B16 melanoma cells and to optimize GANT61 concentration for subsequent experiments, we performed a cell viability assay; a significant decrease in growth was observed in B16 cells stimulated for 48 h with 20 μM of GANT61 compared to the control (Figure 4A).

### 2.4. Comprehensive Gene Expression Analysis

For clarification of the gene expression profile of GANT61 in B16 cells, we treated B16 cells with 20 µM of GANT61 for 48 h (based on the data shown in Figure 4A) and performed a whole-genome microarray analysis of 22,100 gene expression changes. Figure 4B is a scatterplot of the normalized data (normalization method quantiles) for all probes that have revealed gene expression in B16 cells exposed to GANT61. The number of genes with variable expression of GANT61 is 1091 (probes): there are 553 genes with increased expression (i.e., a Z-score ≥ 2.0, ratio ≥ 1.5) and 538 genes with decreased expression (Z-score ≤ −2.0, ratio ≤ 0.6), and signal values below 100 are also shown (Figure 4B).

### 2.5. Identification of Gene Expression Profiles in B16 Cells Exposed to GANT61

Table 1 lists the 10 genes that are most up- or downregulated after the 48 h of GANT61 treatment. The most upregulated genes are methyltransferase-like 22 (Mettl22, Z-score of 6.492), interferon-induced protein with tetratricopeptide repeats 3 (Ifit3, Z-score of 5.261), Ifit3b (Z-score of 4.901), ubiquitin-specific peptidase 18 (Usp18, Z-score of 4.142), D site albumin promoter-binding protein (Dbp, Z-score of 3.937), dCMP deaminase (Dctd, Z-score of 3.900), myeloid cell nuclear differentiation antigen (Mnda, Z-score of 3.838), FXYD domain-containing ion transport regulator 5 (Fxyd5, Z-score of 3.823), RAD51 homolog C (Rad51c, Z-score of 3.698), and DEAD (Asp-Glu-Ala-Asp) box polypeptide 60 (Ddx60, Z-score of 3.691). The most downregulated genes are tetraspanin 10 (Tspan10, Z-score of −6.100), carbonic anhydrase 6 (Car6, Z-score of −4.371), EGF domain-specific O-linked N-acetylglucosamine (GlcNAc) transferase (Eogt, Z-score of −4.134), ctype 13A2 (Atp13a2, Z-score of −4.041), Golgi reassembly-stacking protein 1 (Gorasp1, Z-score of −4.025), DIS3 mitotic control homolog (*S. cerevisiae*)-like (Dis3l, Z-score of −3.994), transmembrane protein 208 (Tmem208, Z-score of −3.919), solute carrier family 12, member 7 (Slc12a7, Z-score of −3.907), perilipin 2 (Plin2, Z-score of −3.773), lysine (K)-specific demethylase 3A (Kdm3a, Z-score of −3.739), and folliculin (Flcn, Z-score of −3.611).

### 2.6. Validation of Microarray Findings with Real-Time PCR

For a validation of the microarray results, a real-time RT-PCR was performed on the 10 genes most downregulated after the 48 h GANT61 treatment with B16 cells. β-actin was used as an endogenous control. The real-time RT-PCR results revealed that the genes TSPAN 10, Car6, Eogt, and Atp13a2 were significantly downregulated after the GANT61 treatment compared to the control group (Figure 4C, *p* < 0.05), whereas Gorasp1, Dis3l, Slc12a7, Plin2, Kdm3a, and Flcn were not significantly downregulated after the exposure to GANT61 (Appendix A). A real-time RT-PCR was also performed using GAPDH as an endogenous control, and significant downregulations of TSPAN 10, Car6, Eogt, and Atp13a2 genes was observed after the GANT61 treatment (*p* < 0.05), whereas the genes Gorasp1, Dis3l, Tmem208, Slc12a7, Plin2, Kdm3a, and Flcn were not significantly decreased after the GANT61 treatment.

### 2.7. Pathway Analysis

We imported the genes with significantly different amounts of expression (Z-score > 2.0) across the various comparisons into the DAVID ver. 6.8 annotation tool and performed GO (gene ontology) and KEGG (Kyoto Encyclopedia of Genes and Genomes) pathway analyses [24]. The initial data were generated using the 48 h GANT61 exposure group results and categorized based on GO terms with *p*-values < 0.05. Table 2 shows the ten most frequently derived GO functional categories obtained using the data from the samples exposed to GANT61 for 48 h: G protein-coupled receptor signaling pathway (gene count *n* = 101), defense response to virus (*n* = 24), negative regulation of viral genome replication (*n* = 10), sensory perception of smell (*n* = 72), spermatogenesis (*n* = 38), protein kinase B signaling (*n* = 8), response to virus (*n* = 11), bicarbonate transport (*n* = 5), response to type 1 interferon (*n* = 4), limb bud formation (*n* = 4), regulation of viral entry into host cell (*n* = 5), and cellular response to interferon-alpha (*n* = 5). The enrichment of specific pathway components into functionally regulated gene groups was characterized with reference to the KEGG pathway database. After the 48 h exposure to GANT61, the major genes identified are those involved in olfactory transduction (gene count *n* = 77), alcoholism (*n* = 17), chemokine signaling pathway (*n* = 16), tyrosine metabolism (*n* = 6), cocaine addiction (*n* = 6), relaxin signaling pathway (*n* = 11), phenylalanine metabolism (*n* = 4), RIG-I-like receptor signaling pathway (*n* = 7), and alanine, aspartate, and glutamate metabolism (*n* = 5) (Table 3). The normal enrichment score (NES), *p*-value, and false discovery rate (FDR) for the ‘Complement and Coagulation Cascade’ gene set all decrease as the GANT61 concentration increases, suggesting that GANT61 induces a dose-dependent downregulation of this pathway.

### 2.8. GSEA Results

We performed a GSEA of the effect of GANT61 on B16 cells. The results demonstrated that genes involved in ‘apoptosis signaling pathway’, ‘sprouting’, ‘angiogenesis’, and ‘PD-L1 expression and PD-1 checkpoint pathway in cancer’ were significantly altered by the GANT61 treatment (Figure 4D, *p* < 0.05).

### 2.9. Proportion of Apoptotic Cells

The GSEA revealed the possibility that GANT61 regulates apoptosis and PD-L1 expression signaling in B16 cells, and thus, we examined the association between apoptosis and PD-L1-expressing cells using flow cytometry. The percentages of cells stained with PI-negative and annexin V-positive were 0.5% in the control group and 0.8% in the GANT61-treated group, indicating that GANT61 treatment increased early apoptosis (Figure 5A, *p* = 0.058). The percentage of PD-L1-positive cells among those cells stained as PI-negative and annexin V-positive was 0.0% in the GANT61-treated group versus 6.7% in the control group, revealing a trend toward decreased PD-L1 expression in cells in which early apoptosis was induced by GANT61 treatment (Figure 5A, *p* = 0.134). In contrast, the percentages of PI-positive and annexin V-positive cells were 11.4% in the control group and 14.2% in the GANT61-treated group, indicating that GANT61 significantly promoted late apoptosis (Figure 5B, *p* = 0.000). The percentage of PD-L1-positive cells among those cells stained as PI-positive and annexin V-positive was 92.8% in the control group and 88.7% in the GANT61-treated group, indicating that (i) GANT61-treated cells underwent late apoptosis, and (ii) PD-L1 expression was significantly decreased in cells in which late apoptosis was induced by the GANT61 treatment (Figure 5B, *p* = 0.000).

## 3. Discussion

Melanoma is a malignancy of melanocytes, which are melanin (pigment)-producing cells in the basal layer of the epidermis. Melanocytes are of neural crest origin and, therefore, express many signaling molecules and factors that promote migration and metastasis after malignant transformation. Advanced cases of oral melanoma often show jaw bone invasion, and the 5-year survival rate for these patients is significantly low [8]. Jaw bone invasion is also observed in gingival cancer patients, whose 5-year survival rate is also significantly reduced, and cancer jaw bone invasion has been demonstrated to be an independent prognostic factor [25], indicating that the cancer bone microenvironment could become a novel target for advanced cancer with jawbone destruction.

To recapitulate this cancer bone microenvironment in vivo in the present study, we used mouse tibiae as they are a basic multicellular unit with important anatomical structures in bone that contribute to bone homeostasis. Numerous growth factors released during normal bone metabolism may promote the growth of B16 cells, but it is known that B16 cells disrupt normal bone homeostasis [26]. The mice used in the present study were 5 weeks old and had growth plates at the tibial epiphysis, suggesting that tumor cells exerted growth in the longitudinal direction of the tibia.

DMSO treatment after B16 cell inoculation reduced the number of TRAP-positive osteoclasts expressed in the ossification zone. We suspect that the normal growth of growth-plate chondrocytes might have been disrupted in the DMSO-treated tumor group compared to the non-tumor group and the GANT61-treated group; however, further experiments are needed to clarify the relationship between the numbers of osteoclasts and the length of each zone of chondrocytes. The present results suggest that the replacement of chondrocytes with bone in the final differentiation process of endochondral ossification might have been inhibited in the DMSO-treated group with B16 cell inoculation.

GANT61 was administered intraperitoneally at a concentration of 40 mg/kg every other day for two weeks, but it had no significant effect on the process of endochondral bone formation other than inhibition of the ossification zone, as indicated by the hematoxylin and eosin (HE)- and TRAP-stained images of the mice’s tibiae. However, Indian hedgehog (Ihh), which activates Gli1 and Gli2, is expressed in prehypertrophic chondrocytes, and it regulates the initiation of hypertrophic chondrocyte differentiation in a negative feedback loop with parathyroid hormone-related protein (PTHrP) [27]. In a PTHrP-independent system, Ihh signaling functions in concert with osteogenic proteins (i.e., bone morphogenetic proteins [BMPs]) to induce the differentiation of progenitor cells to osteoblasts [28]. In light of these previous reports, we speculate that GANT61 may affect the process of endochondral ossification. Further investigations are necessary to clarify the effects of GANT61 on the developmental stage.

G protein-coupled receptors (GPCRs), the largest family of cell-surface molecules involved in signal transmission, play crucial roles in tumor growth and metastasis [29]. In melanoma, activating/inactivating mutations of GPCRs include melanocortin 1 receptor (MC1R) [30]. There are also reports of the following in melanoma: glutamate family of G protein-linked receptors (GRM1–8) [31,32], muscarinic receptor [33], selected GPCR ligands, signaling pathways including Frizzled (Fz) PAR1 parathyroid receptor1 (PTHR1) [34], and chemokine receptor (CXCR4) [35]. In addition, GPCRs are involved in tumor progression via Hh signaling [36,37]. Mutations in SMO, one of the GPCRs, are also present in various tumors [38]. In our previous study, SHH acted directly on osteoclast progenitor cells and stimulated RANKL-induced osteoclastogenesis, and the pathway analysis revealed that most of the pathways involved GPCR-related genes [39]. The direct effect of GANT61 on individual GPCRs in melanoma cells were not evident based on the Z-scores and ratios, which showed no significant changes. However, to reveal a comprehensive paradigm of GANT61 function, gene ontology (GO) functional category analysis revealed that the GRCR signaling pathways were significantly altered by the GANT61 treatment (Table 2, *p* = 0.000051).

Olfactory receptors are the largest gene family in humans, with 408 active coding genes and over 600 pseudogenes reported to date [40]. Olfactory receptors have a role in the survival of patients with invasive breast cancer and may serve as prognostic indicators [41]. Olfactory receptors also control the growth and migration of human melanomas [42,43], and Hh signaling controls olfactory function and the KEGG pathway [44]. In the present study, GANT61 treatment produced a significant shift in olfactory transduction (Table 3, *p* = 0.00037), suggesting that olfactory transduction is involved in melanoma proliferation and invasion in the bone microenvironment. Further investigations using patient samples are needed to investigate this possibility.

Tumor cells evade immune surveillance by upregulating the surface expression of PD-L1, which interacts with PD-1 on T cells to trigger immune checkpoint responses [45,46]. Anti-PD-1 antibodies have shown remarkable promise in the treatment of metastatic melanoma [47]; however, the response rate among patients is still low [48,49]. In support of these findings, a study using pancreatic ductal adenocarcinoma cells and gastric cancer cells revealed cell lines that showed Hh signaling-induced PD-L1 expression [50], and increased Hh activity correlated with multiple immunosuppressive characteristics in the tumor microenvironment of diverse cancers [51]. We determined the direct effect of GANT61 on PD-L1 gene expression in melanoma cells using *p*-values or log2 fold-change from the differential expression results of the GSEA to identify whether gene sets work together in a coordinated manner. Although no significant change in PD-L1 expression after GANT61 treatment was observed based on the Z-scores and ratios, the GSEA revealed that many genes with low fold-change after GANT61 treatment were found to have a significant effect by working in concert with PD-L1 expression and the PD-1 checkpoint pathway in melanoma.

Intrinsic induction of PD-L1 occurs via several oncogenic and transcriptional pathways involved in the non-canonical mechanisms of Gli activation [52]. Our present findings suggest that the inhibitory effects of GANT61 on melanoma in the bone microenvironment include (i) an induction of late apoptosis and (ii) a suppression of PD-L1 expression. It was reported that an inhibition of PD-L1 with an anti-PD-L1 antibody or the Hh inhibitor cyclopamine enhanced lymphocyte antitumor activity in Panc-1 cells co-cultured with lymphocytes [50]. In a mouse model of gastric cancer, GANT61 suppressed tumor growth via decreased PD-L1 expression, which was accompanied by an increased number of CD8+ cytotoxic T lymphocytes [53]. It was recently reported that the mTOR pathway mediated the non-canonical Hh signaling cascade to induce PD-L1 expression [54], and that increased HH activity was a predictive biomarker for resistance to immune checkpoint inhibitors in diverse cancers [51]. Combinatorial drug therapy with Hh signal transduction and immune checkpoint inhibition may be appropriate for eligible melanoma patients with jaw bone destruction, although further studies are necessary to test this speculation.

Tumor vascularization supplies oxygen and nutrients for tumor cells’ proliferation, but these vessels may be used to contribute to an inhibition of tumor growth via a recruitment of immune cells and for drug delivery [55]. The results of the present GSEA and in vivo analysis revealed that the GANT61 treatment significantly suppressed the formation of a tumor vascular network in the cancer bone microenvironment. Endomucin is a type I integral membrane glycoprotein expressed apically by capillary and venous endothelial cells, and it accounts for most of the vascular networks formed in melanoma tumor masses [56]. However, it is unclear whether abnormal angiogenesis in the melanoma bone microenvironment is associated with endomucin expression levels via Hh signaling. PTCH1-positive tumor blood vessels were found adjacent to tumor cells that strongly expressed SHH in a previous study (Int. J. Mol. Sci. 2019, 20(22), 5779; https://doi.org/10.3390/ijms20225779), and Gli1 and Gli2 were highly expressed in the tumor vasculature in the melanoma bone microenvironment in the present study (Figure 1D). However, the direct effects of GANT61 on SHH, Gli1, and Gli2 expressions in melanoma cells were not evident, showing no significant change unlike the results of the in vivo analysis. Tumor cells need large amounts of nutrients and growth factors, which are supplied and distributed to tumor tissues by aberrant tumor vasculature [57]. It is suggested that growth factors, including SHH, produced by tumor cells paracrinely induce Gli1 and Gli2 expressions in tumor vascular endothelial cells, and various angiocline factors produced by tumor vascular endothelial cells induce SHH and Gli1 in tumor cells (Figure 6). In the melanoma bone microenvironment, mutual crosstalk between tumor cells, tumor vascular endothelial cells, and osteoclasts is implicated in the expression of Hh signaling in tumor cells [58,59].

MECA-79+ tumor-associated high endothelial venules (HEVs) are frequently present in human tumors and have been proposed to play important roles in lymphocytes’ entry into tumors in a process that is essential for successful antitumor immunity and lymphocyte-mediated cancer immunotherapy with immune checkpoint inhibitors [60]. HEVs have been observed in human melanomas, and the density of HEVs in melanomas correlates with the density of CD3+ and CD8+ T cells and favorable clinical parameters [61,62]. However, the variation in the number of HEV cells involved in lymphocytic infiltration of melanoma in the bone microenvironment and their localization with endomucin-positive cells is not clear and is a subject for future studies.

Ten years after the approval of immune checkpoint inhibitors for advanced melanoma, it is time to reflect on the lessons learned about immune system regulation in cancer treatment and consider new approaches to therapy [63]. One of the current challenges in melanoma antitumor immunotherapy is immunosuppression in the tumor microenvironment [63,64], and it is, thus, critical to promote remodeling of the tumor bone microenvironment, normalization of abnormal angiogenesis, and immune cell infiltration in cancer in order to enhance antitumor effects on advanced melanoma with jaw bone invasion (Figure 6).

## 4. Materials and Methods

All methods involving humans and animals in this study were performed in accordance with relevant guidelines and regulations.

### 4.1. Tissue Array Analysis

A human melanoma tissue array was purchased from U.S. Biomax (Rockville, MD, USA; cat. no. ME481c 228, 40 cases of malignant melanoma and 8 samples of normal skin tissue). The tissue array slides were stored at 4 °C; they might not have been fresh cut but were suitable for immunohistochemistry. The following antibodies were used for the immunohistochemical analysis of the tissue array: SHH (1:500, #ab53281, rabbit IgG, Abcam, Cambridge, UK), Gli1 (1:1000, #NBP1-78259, rabbit IgG, Novus Biologicals, Englewood, CO, USA), and Gli2 (1:150, #NB600-874SS, rabbit IgG, Novus Biologicals) antibodies. Immunostained sections were then counterstained with methylene green, observed, and photographed with a light microscope (Nikon, Tokyo, Japan). An immunohistochemical analysis was performed to identify any changes in the numbers of cells in the human melanoma tissue array that were positive for SHH, Gli1, and Gli2. The images of five random fields from each sample (20× objective) were taken, and the number of positive cells per mm^2^ was calculated. The staining intensity (SI) was evaluated visually, and each specimen was assigned to one of the following categories: negative (0), weak intensity (1), moderate intensity (2), or strong intensity (3). There were 3 samples with tissue detached, and the number of samples analyzed was *n* = 37.

### 4.2. Patients and Samples

Patient samples were obtained from the Oral Pathology Department of Okayama University. This study was approved by the Ethics Committee of Okayama University Graduate School of Medicine, Dentistry and Pharmaceutical Sciences (project identification code: 1608–018; date of approval: 10 March 2017; name of the ethics committee: Analysis of Biological Properties of Oral Cancer). Informed consent was obtained from all subjects. None of the patients had received chemotherapy, radiotherapy, or immunotherapy before undergoing the sampling.

### 4.3. Cell Lines and Culture Conditions

The B16 mouse melanoma cell line was purchased from the RIKEN Cell Bank (Tsukuba Science City, Japan). B16 cells were cultured at a density of 1 × 10^5^ cells/cm^2^ in a high-glucose Dulbecco’s modified Eagle’s medium (DMEM) (Sigma-Aldrich, St. Louis, MO, USA) containing 5% fetal bovine serum (FBS) (Biowest, Logan, UT, USA), followed by replacement with DMEM/F12 containing 5% FBS, 10 µg/mL of L-glutamine (Life Technologies, Grand Island, NY, USA), and 10 µg/mL of antibiotic-antimycotic solution (Life Technologies). The cells were then cultured at 37 °C for various periods up to 12 days under 5% CO_2_. The cell line was characterized by genotyping at the Cell Bank.

### 4.4. Animal Study

A mouse model of bone destruction was prepared by inoculating 5-week-old female C57BL mice under general anesthesia (intraperitoneal (IP) injection of 0.15 mg/kg of medetomidine, 2.0 mg/kg of midazolam and 2.5 mg/kg of butorphanol) with cell suspensions of B16 mouse melanoma cells (1 × 10^5^/10 µL of phosphate-buffered saline [PBS]) via injection into the bone marrow space of the right tibial metaphysis. The mice were then randomly assigned to two groups (*n* = 8/group). After the B16 cell inoculation, the mice were intraperitoneally administered GANT61 (HY-13901, MedChemExpress, Monmouth Junction, NJ) (40 mg/kg) or dimethyl sulfoxide (DMSO) (cat. no. 046-21981, Fujifilm Wako Pure Chemical Industries, Osaka, Japan) as a vehicle on alternating days from day 3 to day 15. On day 16, all of the mice were euthanized with 150 mg/kg of pentobarbital via IP administration, and the tibiae were removed. The protocols for the mice were approved by the Ethics Review Committee for Animal Experimentation of the Health Sciences University of Hokkaido, Graduate School of Dentistry and Pharmaceutical Sciences (ethical permission code: 19-088). The study was carried out in compliance with the ARRIVE guidelines.

### 4.5. In Vivo Radiography and Measurement of Osteolytic Lesion Areas

Tibiae were excised from the euthanized mice with enucleation of the tumor, and bones were dissected free of tissue and fixed in a 4% paraformaldehyde solution. The fixed bones were scanned using a computed tomography (CT) system (Veraview X800, J. Morita Mfg., Kyoto, Japan) at 60–100 kV with a detection voxel size of 80 μm and 2.5 LP/mm resolution. The scanned images were reconstructed with the use of DICOM viewer software (OsiriX-N-20.50, Sapporo, Japan, Newton Graphics, Inc.)

### 4.6. Immunohistochemistry

For further assessment of mouse bone invasion by B16 mouse melanoma cells, the tibiae were excised and fixed in 4% paraformaldehyde in 0.1 M of PBS (pH 7.4) for 24 h at 4 °C. The specimens were decalcified by 10% ethylenediaminetetraacetic acid (EDTA, pH 7.4, Hayashi Pure Chemical Industries, Osaka, Japan) for 14 days at room temperature and then embedded in paraffin (Fisher Scientific, Fair Lawn, NJ, USA). Thin serial sections (4.5 µm) were cut longitudinally, and the sections were deparaffinized with xylene, rehydrated with ethanol, and stained with Mayer’s hematoxylin and eosin (H&E) solution (Muto Pure Chemicals Co., Tokyo, Japan).

For immunohistochemistry, dehydrated sections of bone invasion were treated with 0.3% H_2_O_2_ in the PBS (pH 7.4) for 30 min at room temperature to inactivate endogenous peroxidase. The sections were pretreated with 3% bovine serum albumin (BSA) (Sigma-Aldrich, St. Louis, MO, USA) in the PBS for 30 min at room temperature, followed by incubation with primary antibodies against Melan-A (1:1000, #18472-AP, rabbit IgG, Proteintech, Rosemont, IL, USA), proliferating cell nuclear antigen (PCNA) (1:1000, #ab18197, rabbit IgG, Abcam, Cambridge, MA, USA), Shh (1:500, #ab53281, rabbit IgG, Abcam), Gli1 (1:1000, #NBP1-78259, rabbit IgG, Novus Biologicals, Centennial, CO, USA), Gli2 (1:150, #NB600-874SS, rabbit IgG, Novus Biologicals), PDL-1/CD274 (1:500, #17952-1-AP, rabbit IgG, Proteintech), and endomucin V.7C7 (1:100, #sc65495, rat IgG, Santa Cruz Biotechnology, Santa Cruz, CA, USA) overnight at 4 °C.

The sections were then reacted with Histofine Simple Stain mouse MAX-PO (Rabbit 414341F, Rat 414311F; Nichirei, Tokyo, Japan) for 1 h at room temperature. Color was developed with the use of a liquid diaminobenzidine substrate-chromogen system (Dako, Carpinteria, CA, USA). The immunostained sections were counterstained with 1% methylene green (Muto Pure Chemicals). An immunohistochemical analysis was performed to quantify any changes in the number of positive cells of Melan-A in the bone marrow of the tibiae. The images of five random fields of bone marrow from each sample (20× objective) were taken, and the number of positive cells per mm^2^ was calculated. For the determination of the LI, the percentage of positive cells in Melan-A-positive tumor cells/mm^2^ was observed at ×400 magnification and calculated as the LI.

### 4.7. TRAP Staining

Formalin-fixed paraffin-embedded sections were cut at 5 µm, deparaffinized with xylene, and rehydrated with ethanol. Tartrate-resistant acid phosphatase (TRAP) staining was performed using a TRAP kit (Sigma-Aldrich). The number of TRAP-positive osteoclasts per millimeter of tumor–bone interface was counted at 20× magnification of each sample.

### 4.8. Cell Viability Assays

Cell viability was determined using the cell proliferation reagent WST-1 (Sigma-Aldrich). B16 melanoma cells were seeded at 5 × 10^3^/well in 96-well plates in DMEM containing 10% FBS and cultured overnight under 5% CO_2_. The cells were then treated with a range of GANT61 concentrations (0, 1.25, 2.5, 5, 10, and 20 µM) dissolved in DMSO. Following incubation for 24 or 48 h, 10 µL of WST-1 was added to each well, and the cells were cultured for 1 h. The absorbance at 450 nm was determined using a microplate reader (Model 680, Bio-Rad Laboratories, Hercules, CA, USA).

### 4.9. Gene Expression Microarrays

Total RNA was isolated from the cells with the use of an RNeasy Mini Kit (Qiagen, Hilden, Germany) according to the manufacturer’s instructions, and the RNA samples were quantified using a spectrophotometer (ND-1000, NanoDrop Technologies, Wilmington, DE, USA). The quality of the RNA samples was confirmed with a 2200 TapeStation (Agilent Technologies, Santa Clara, CA, USA). cRNA was amplified, labeled with total RNA with the use of a GeneChip™ WT PLUS Reagent Kit (Thermo Fisher Scientific, Waltham, MA, USA), and hybridized using a Thermo Fisher Scientific Clariom™ S Assay (rat) according to the manufacturer’s instructions. All hybridized microarrays were scanned using an Affymetrix scanner (Thermo Fisher), and the relative hybridization intensities and background hybridization values were calculated using a Thermo Fisher Expression Console™.

### 4.10. Data Analysis and Filter Criteria

The raw signal intensity for each probe was calculated from the hybridization intensities, and the raw signal intensities of two samples were log2-transformed and normalized using the signal space transformation-robust multiple-array average normalization (SST-RMA) method and quantile algorithm [65] with the Thermo Fisher Expression Console™ 1.1 software (Thermo Fisher Scientific). To identify up- and downregulated genes, we calculated Z-scores [66] and ratios (non-log scaled fold-change) from the normalized signal intensity of each probe for comparison between the control and experimental samples. The following criteria for regulated genes were then established: Upregulated genes: Z-score ≥ 2.0 and ratio ≥ 1.5-fold, and downregulated genes: Z-score ≤ −2.0 and ratio ≤ 0.66. The enrichment of specific pathway components in functionally regulated gene groups was characterized with reference to the Gene Ontology (GO) database and the Kyoto Encyclopedia of Genes and Genomes (KEGG) pathway database. We used the tools and data provided by the Database for Annotation, Visualization and Integrated Discovery (DAVID) (http://david.abcc.ncifcrf.gov/home.jsp, accessed on 16 April 2022) [67].

### 4.11. Gene Set Enrichment Analysis (GSEA)

The differentially expressed transcripts obtained through RNA-seq after 48 h treatment with GANT61 were compared to the curated gene sets from Gene Ontology Biological Processes (GOBP) with the use of the gene set enrichment analysis (GSEA) tool (https://www.gsea-msigdb.org/gsea/index.jsp, accessed on 7 December 2021) [68]. The results of this GSEA were evaluated based on the enrichment score (ES), which represents the extent to which a gene set is over-represented at the top or bottom of a ranked list of genes of interest. The normalized enrichment score (NES) accounts for the difference in the gene set size and for the correlations between the gene set and the expression dataset. Other values obtained in this analysis were the nominal *p*-value, which represents the significance of the enrichment score, and the false discovery rate (FDR), which indicates the probability that the results represent a false positive finding [68].

### 4.12. Real-Time Reverse Transcription-Polymerase Chain Reaction (RT-PCR)

Total RNA was isolated from B16 cells by using the TRIZOL reagent (Life Technologies, Rockville, MD, USA) according to the manufacturer’s recommendations. Reverse transcription of the extracted RNA was performed using the ReverTra Ace^®^ qPCR RT Master Mix (Toyobo, Osaka, Japan). A real-time RT-PCR was performed to determine the expression level of mRNA using a LightCycler^®^ Nano instrument (Roche Diagnostics, Basel, Switzerland). The reaction mix for the PCR consisted of cDNA, a pair of primers, and KAPA SYBR FAST qPCR Mix (Nippon Genetics, Tokyo, Japan). The PCR was performed under the following conditions: initial incubation at 50 °C for 2 min, denaturation at 95 °C for 10 min, 40 cycles of denaturation at 95 °C for 15 s, and annulation at 60 °C for 1 min. The relative mRNA expression level was calculated as the quantification cycle (Cq) value obtained after subtracting the Cq value of glyceraldehyde-3-phosphate dehydrogenase (GAPDH) from the Cq value of the target gene, using the ∆∆Cq method. The primer sequences used in this study are shown in Appendix A.

### 4.13. Flow Cytometry

The effect of the Hh enzyme inhibitor GANT61 on apoptosis in the cultured melanoma cells was investigated using flow cytometry. After melanoma cells treated with/without GANT61 (GANT61/control) were cultured, the cells were harvested from each cell system and filtered using a Falcon Cell Strainer (100 μm; Corning, New York, NY, USA). The cells were incubated in the dark for 30 min at 4°C with fluorescein isothiocyanate (FITC) (Alexa Fluor 488), an Annexin V Apoptosis Detection Kit I (BD Biosciences, Franklin Lakes, NJ, USA), and PE Rat Anti-Mouse CD274 (programmed cell death 1 ligand 1 [PD-L1], BD Biosciences). Propidium iodide (PI), annexin V (Alexa Fluor 488), and PD-L1 (PE) were analyzed using a flow cytometer (BD FACSAriaⅢu, BD Biosciences). A total of 10,000 events was evaluated by the flow cytometer. Each population was hierarchically linked as P4 to P5 to P6 of gating. Early apoptosis was defined as PI-negative and annexin V-positive, and late apoptosis was defined as PI-positive and annexin V-positive. Furthermore, PD-L1-positive cells were counted among those cells via gating [69,70].

### 4.14. Statistical Analysis

The results were analyzed using unpaired Student’s t-test for the comparisons of two groups and one-way analysis of variance (ANOVA) for the analysis of multiple group comparisons. Cell viability was determined based on Dunnett’s test. In the flow cytometric analysis, χ^2^-test was used, and results with a *p*-value < 0.05 were considered significant. The GSEA results were analyzed using Fisher’s exact test. All experiments were performed in quadruplicate. Differences were considered significant when *p* < 0.05. All analyses were performed using SPSS ver. 26 (IBM, New York, NY, USA). The Spearman correlations between intensities were analyzed using GraphPad Prism 6.0 (GraphPad Software, San Diego, CA, USA).

## Figures and Tables

**Figure 1 ijms-24-08862-f001:**
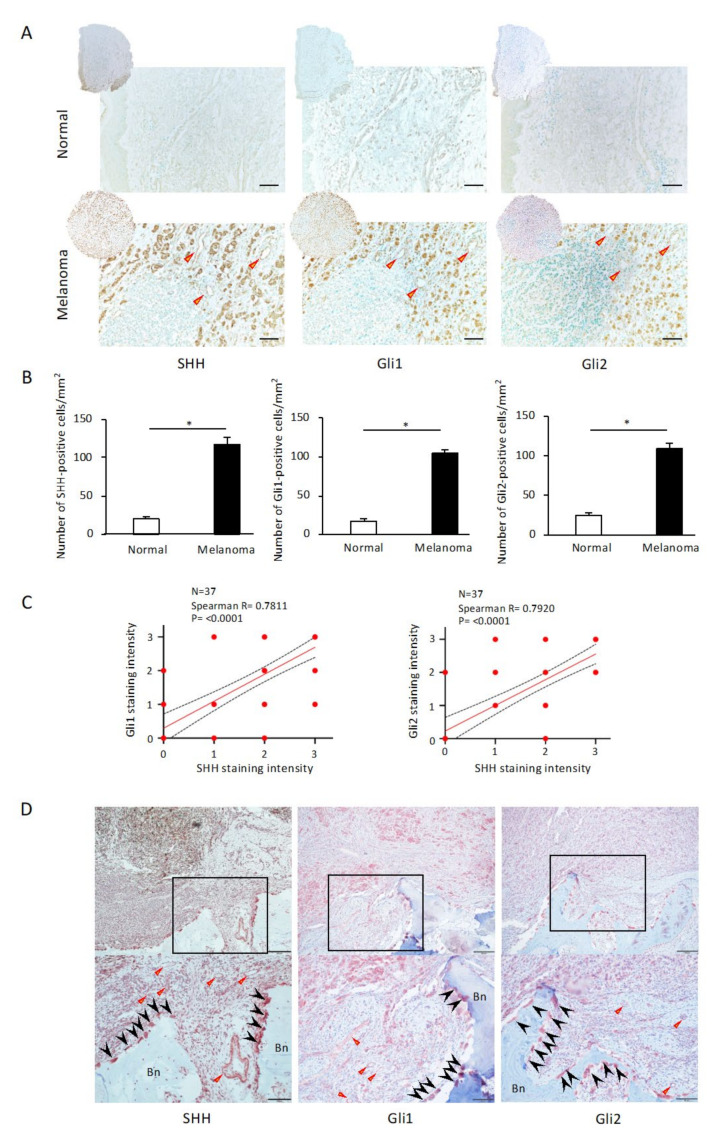
Immunohistochemical analysis of the expressions of Sonic Hedgehog (SHH), Gli1, and Gli2 in human melanoma samples. (**A**) Representative images of SHH, Gli1, and Gli2 expressions in normal dermal tissue and melanoma. SHH, Gli1, and Gli2 are expressed not only in melanoma cells but also in tumor vascular endothelial cells in the stroma. Scale bar: 200 µm. Arrowhead: tumor vasculature. (**B**) The numbers of SHH-, Gli1-, and Gli2-positive cells/mm^2^ are significantly higher in melanoma tissues than in normal skin tissues. The data from a typical experiment (mean ± SD) are presented. * *p* < 0.05 between the indicated groups. (**C**) Staining intensity (SI, A) is evaluated visually: negative (0), weak (1), moderate (2), and strong (3). SI3 (SHH and Gli1, *n* = 15), SI3 (SHH and Gli2, *n* = 12), SI2 (SHH and Gli1, *n* = 7), and SI2 (SHH and Gli2, *n* = 4). Spearman’s correlations between the intensities of SHH and Gli1 (left) or Gli2 (right) were analyzed using GraphPad Prism 6.0. (**D**) Immunohistochemical staining for SHH, Gli1, and Gli2 in osteolytic malignant melanoma of the maxilla. Each photo is a magnification of the rectangle-delimited area corresponding to a melanoma bone-destructive area. Scale bars: 200 µm (upper) and 100 µm (lower). Arrowhead*:* osteoclasts. Triangular arrowheads: tumor vasculature. Bn: bone.

**Figure 2 ijms-24-08862-f002:**
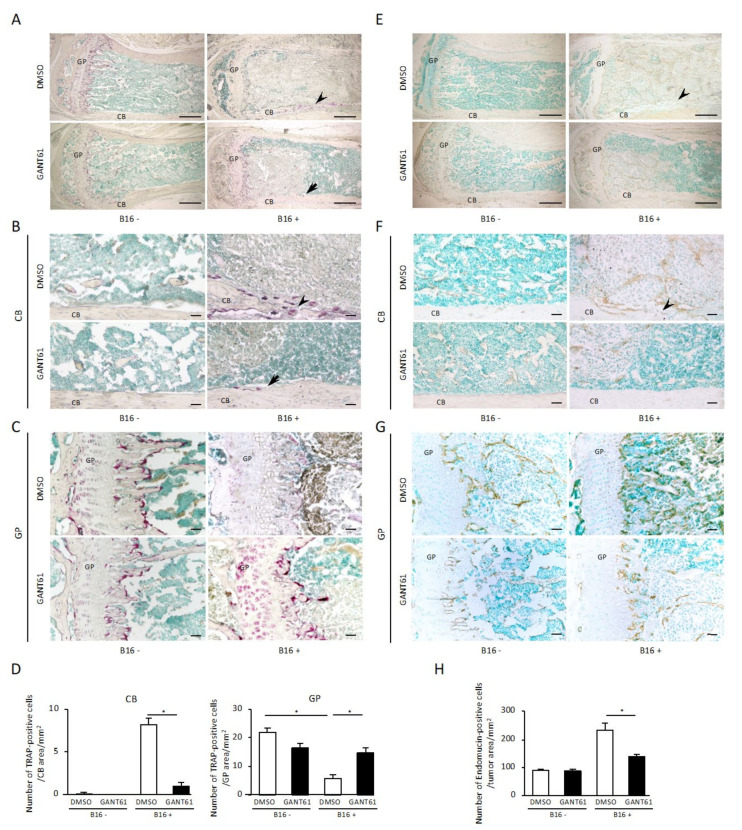
Histochemical analysis for TRAP and endomucin staining of tibial metaphyses bearing B16 cells. (**A**–**C**,**E**–**G**) Representative images of TRAP (**A**–**C**) and endomucin (**E**–**G**) staining of tibial metaphyses with (right) or without (left) B16 cell inoculation and treated with DMSO (upper) or GANT61 (lower). Each image is a magnification of the typical cortical bone area (**B**,**F**) and growth plate area (**C**,**G**) of the corresponding image in panels (**A**,**E**). (**D**) Quantification of TRAP (+) osteoclasts along the bone–tumor interface at the cortical bone area (left) and growth plate area (right) in each experimental group (*n* = 5). (**H**) The number of endomucin positive cells/mm^2^ in tumor tissue area (*n* = 5). Arrowhead: bone rupture. Arrow: bone resorption. CB: cortical bone. GP: growth plate. Data are presented as mean ± SD. * *p* < 0.05 between the indicated groups. Scale bar: 500 µm (**A**,**E**). Scale bar: 50 µm (**B**,**C**,**F**,**G**).

**Figure 3 ijms-24-08862-f003:**
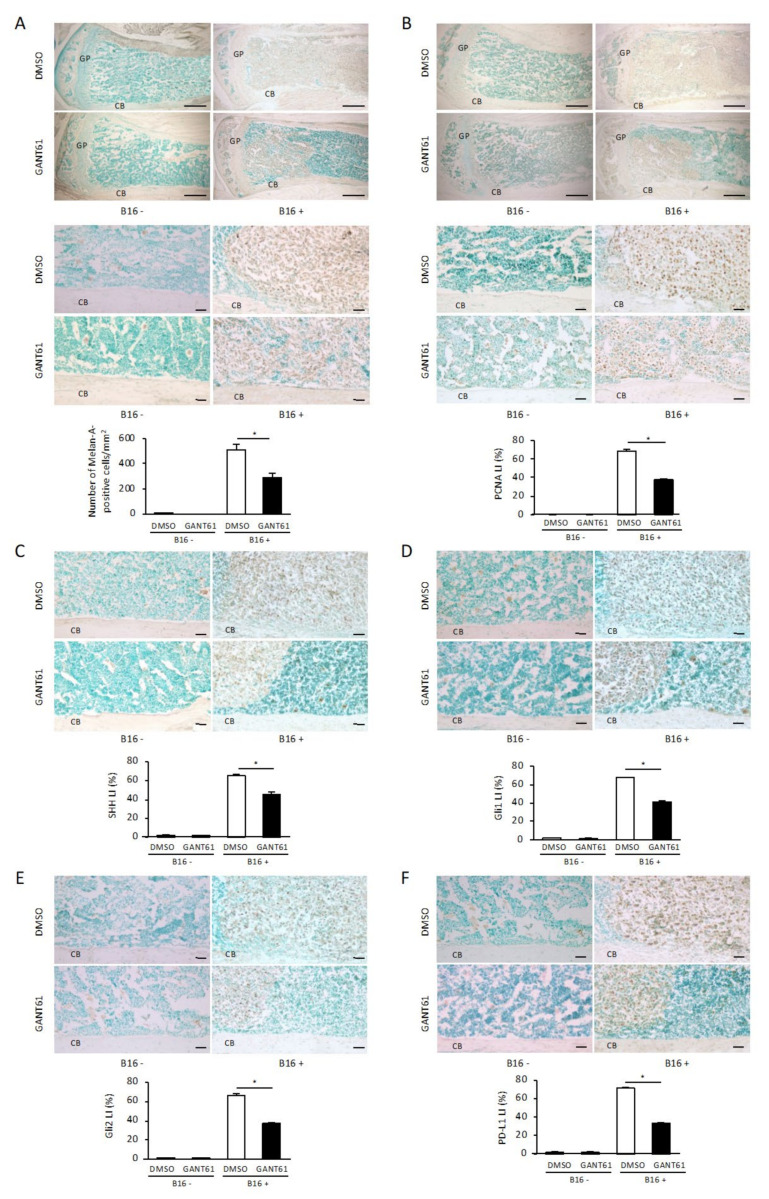
Immunohistochemical analysis of Melan-A, proliferating cell nuclear antigen (PCNA), SHH, Gli1, Gli2, and PD-L1 in bone from mice bearing B16 melanoma cells. (**A**,**B**) Melan-A (**A**) and PCNA (**B**) expression in the tibial metaphysis with (right) or without (left) B16 cell inoculation and treated with DMSO (upper) or GANT61 (lower). Each image is a magnification of the typical cortical bone area (lower parts) of the corresponding upper part images in panels (**A**,**B**). Scale bar: 500 µm (upper part). Scale bar: 50 µm (lower part). The number of Melan-A-positive tumor cells per mm^2^ ((**A**), graph, *n* = 5/group). The percentage of PCNA-positive cells/Melan-A-positive tumor cells per mm^2^ (*n* = 5). (**C**–**F**) The expressions of SHH (**C**), Gli1 (**D**), Gli2 (**E**), and PD-L1 (**F**) in a typical cortical bone area in tibial metaphyses with (right) or without (left) B16 cell inoculation and treated with DMSO (upper) or GANT61 (lower). Scale bar: 50 µm. The percentages of SHH (**C**), Gli1 (**D**), Gli2 (**E**), and PD-L1 (**F**)-positive cells/Melan-A-positive tumor cells per mm^2^ (*n* = 5). Data are presented as mean ± SD. CB: cortical bone. LI: labeling index. * *p* < 0.05 between the indicated groups.

**Figure 4 ijms-24-08862-f004:**
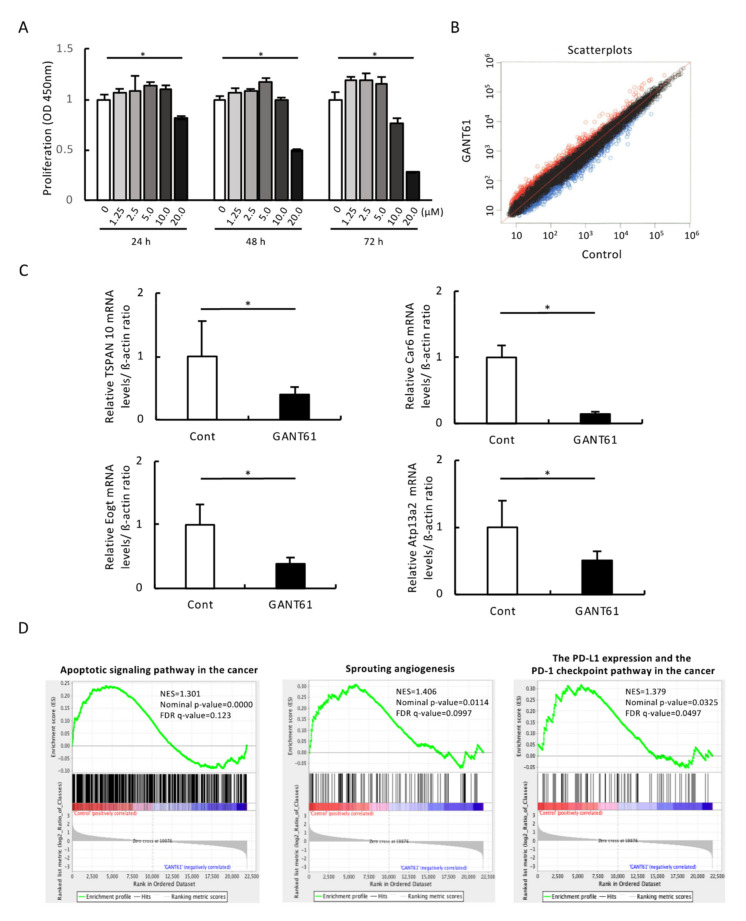
In vitro characterization of GANT61’s effects on B16 cells. (**A**) The effect of GANT61 on the proliferation of mouse B16 melanoma cells. Cell proliferation was quantified using a microplate reader measuring the absorbance of the dye solution at 450 nm (*n* = 4). * *p* < 0.05 between the indicated groups. (**B**) Scatterplots representing the expression of genes in B16 cells exposed to 20 µM of GANT61 for 48 h. *x*-axis: the relative normalized log2-signal intensity of the control (not exposed to GANT61) samples. *y*-axis: the normalized log2-signal intensity of the samples exposed to GANT61. (**C**) Expression changes in the 4 genes that are most downregulated by exposure to GANT61. The genes TSPAN 10, Car6, Eogt, and Atp13a2 were analyzed using real-time RT-PCR (*n* = 5, * *p* < 0.05). β-actin was used as an endogenous control in these protocols. (**D**) The gene ontology biological process (GOBP) results of the gene set enrichment analysis (GSEA) of GANT61-treated B16 melanoma cells. Apoptotic signaling pathway in the cancer—GSEA statistics: normal enrichment score (NES) of 1.301, nominal *p*-value of 0.0000, and FDR q-value of 0.123. Sprouting angiogenesis— GSEA statistics: NES of 1.406, nominal *p*-value of 0.0114, and FDR q-value of 0.0997. The PD-L1 expression and the PD-1 checkpoint pathway in the cancer—GSEA statistics: NES of 1.379, nominal *p*-value of 0.0325, and FDR q-value of 0.0497.

**Figure 5 ijms-24-08862-f005:**
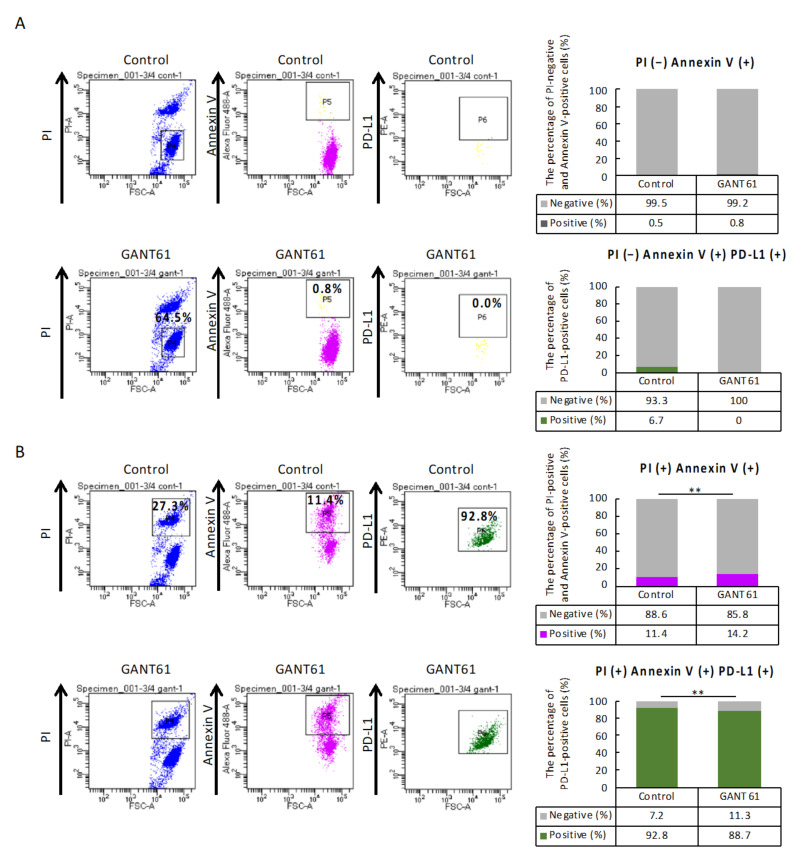
Flow cytometry results of annexin-V and propidium iodide (PI) staining of apoptotic cells following the GANT61 treatment of B16 cells. Each population was hierarchically linked as P4 to P5 to P6 of gating. Early apoptosis was defined as PI-negative and annexin V-positive, and late apoptosis was defined as PI-positive and annexin V-positive. Furthermore, PD-L1-positive cells were counted among those cells via gating. (**A**) Early apoptosis: no significant difference in the number of cells stained as PI- negative and annexin V-positive is observed between the control and GANT61 groups (χ^2^-test, *p* = 0.058, *n* = 4). No significant difference in the number of cells stained with PD-L1-positive among early apoptotic cells is detected between the control and GANT61 groups (χ^2^-test, *p* = 0.134, *n* = 4). (**B**) Late apoptosis: a significant increase in the number of cells stained as PI-positive and annexin V-positive is observed in the GANT61 group compared to the control group (χ^2^-test, ** *p* = 0.000, *n* = 4). A significant decrease in the number of cells stained with PD-L1-positive among late apoptotic cells is detected in the GANT61-treated group compared to the control group (χ^2^-test, ** *p* = 0.000, *n* = 4).

**Figure 6 ijms-24-08862-f006:**
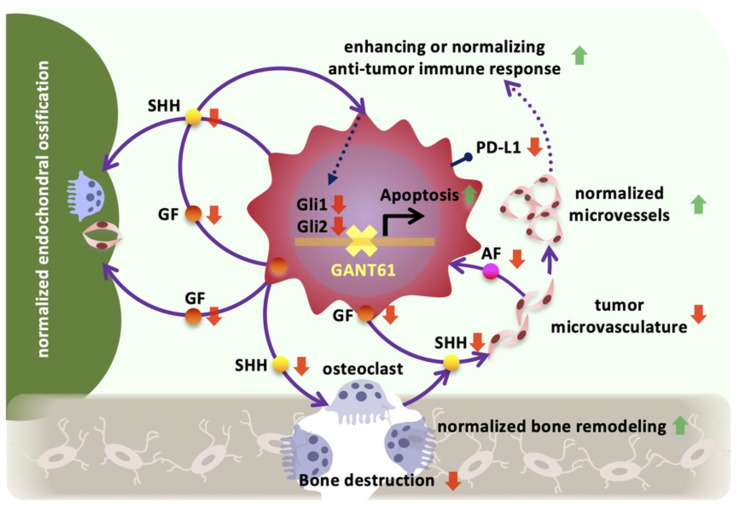
Normalization and remodeling of the tumor bone microenvironment by GANT61. SHH: Sonic Hedgehog; GF: growth factors; AF: angiocrine factors. Red arrows: upregulation. Green arrows: downregulation.

**Table 1 ijms-24-08862-t001:** The 10 most up- and downregulated genes observed among the total of 1091 genes.

Gene Symbol	Gene Description	Z-Score	Ratio
Mettl22	methyltransferase-like 22	6.492	9.517
Ifit3	interferon-induced protein with tetratricopeptide repeats 3	5.261	6.123
Ifit3b	interferon-induced protein with tetratricopeptide repeats 3B	4.901	5.382
Usp18	Ubiquitin-specific peptidase 18	4.142	5.149
Dbp	D site albumin promoter-binding protein	3.937	3.811
Dctd	dCMP deaminase	3.900	3.127
Mnda	myeloid cell nuclear differentiation antigen	3.838	4.528
Fxyd5	FXYD domain-containing ion transport regulator 5	3.823	4.499
Rad51c	RAD51 homolog C	3.698	2.940
Ddx60	DEAD (Asp-Glu-Ala-Asp) box polypeptide 60	3.691	4.256
Flcn	folliculin	−3.611	0.195
Kdm3a	lysine (K)-specific demethylase 3A	−3.739	0.243
Plin2	perilipin 2	−3.773	0.240
Slc12a7	solute carrier family 12, member 7	−3.907	0.172
Tmem208	transmembrane protein 208	−3.919	0.153
Dis3l	DIS3 mitotic control homolog (*S. cerevisiae*)-like	−3.994	0.233
Gorasp1	Golgi reassembly-stacking protein 1	−4.025	0.230
Atp13a2	ctype 13A2	−4.041	0.163
Eogt	EGF domain-specific O-linked N-acetylglucosamine (GlcNAc) transferase	−4.134	0.228
Car6	carbonic anhydrase 6	−4.371	0.194
Tspan10	tetraspanin 10	−6.100	0.104

**Table 2 ijms-24-08862-t002:** The gene ontology (GO) functional category analysis of genes differentially expressed in murine B16 melanoma cells in response to 48 h of GANT61 or control treatment.

Term	Gene Count	*p*-Value
G protein-coupled receptor signaling pathway	101	0.000051
Defense response to virus	24	0.000061
Negative regulation of viral genome replication	10	0.00019
Sensory perception of smell	72	0.00025
Spermatogenesis	38	0.00036
Protein kinase B signaling	8	0.0024
Response to virus	11	0.0038
Bicarbonate transport	5	0.0052
Response to type I interferon	4	0.0086
Limb bud formation	4	0.0086
Regulation of viral entry into host cell	5	0.0093
Cellular response to interferon-alpha	5	0.0093

DAVID v6.8 functional annotation bioinformatics microarray analysis was used to obtain the GO biological process functional categories. Only GO terms for categories that show statistically significantly differences in the number of genes (compared with control) are shown (thresholds: count ≧ 2, *p* < 0.01).

**Table 3 ijms-24-08862-t003:** KEGG pathway functional classification of genes differentially expressed in B16 cells in response to the control treatment and GANT61 for 48 h.

Term	Gene Count	*p*-Value
Olfactory transduction	77	0.00037
Alcoholism	17	0.021
Chemokine signaling pathway	16	0.025
Tyrosine metabolism	6	0.032
Cocaine addiction	6	0.062
Relaxin signaling pathway	11	0.063
Phenylalanine metabolism	4	0.065
RIG-I-like receptor signaling pathway	7	0.092
Alanine, aspartate, and glutamate metabolism	5	0.095

DAVID v6.8 functional annotation bioinformatics microarray analysis was used to obtain the KEGG pathway functional classifications. Only KEGG pathway terms for classifications that show statistically significantly differences in the number of genes (compared with control) are shown (thresholds: count ≧ 2, *p* < 0.1).

## Data Availability

All relevant data are available from the corresponding authors upon reasonable request.

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
