# Peer review of "The Role of Hedgehog Signaling in the Melanoma Tumor Bone Microenvironment"

_ijms, 2023, doi:10.3390/ijms24108862_

Round 1
Reviewer 1 Report
The authors have presented a significant research study suggesting the association of Hh signaling in the mela- 37 noma bone microenvironment through normalization and remodeling of the tumor bone microenvironment. The article is very interesting and has wider research attention. The introduction section is well elaborated with sufficient and appropriate citations. One of the current challenges in melanoma antitumor immunotherapy is immunosuppression by the tumor microenvironment, and it is thus critical to promote remodeling of the tumor bone microenvironment, normalization of abnormal angiogenesis, and immune cell infiltration in cancer in order to enhance the antitumor effects on advanced 500 melanoma with jaw bone invasion. The discussion part is very well summarized and has clearly explained the motive behind writing this research article. I am highly impressed with the discussion of the research article. The summarised diagram is also very much significant in the manuscript.
However, I have some concerns before recommending it for publication:
1. The abstract is not well framed. Kindly elaborate on the conclusion part of the abstract. Reframe lines 31-33.
2. Figure 1 font is different than the font mentioned in the caption. Also, the diagram's quality is not appropriate. Kindly enhance the quality of the diagrams.
3. Figure 2 font is different than the font mentioned in the caption. Also, the diagram's quality is not appropriate. Kindly enhance the quality of the diagrams.
4. I would also suggest the author write a small conclusion deciphering the significance of their research in the field of medicine and health.
Author Response
Dear Reviewer 1
We greatly appreciate the reviewer’s insightful comments regarding our manuscript and are thankful for your considering our paper acceptable in accordance with the reviewers' comments. We have answered all of the reviewers’ comments below, modified the text, added reference and changed Figures accordingly.
Comment #1: The abstract is not well framed. Kindly elaborate on the conclusion part of the abstract. Reframe lines 31-33.
Response #1: As suggested by reviewer, we have changed the text.A crucial regulator in melanoma progression and treatment resistance is the tumor microenvironment, and Hedgehog (Hh) signals activated in a tumor bone microenvironment are a potential new therapeutic target. The mechanism of bone destruction by melanomas involving Hh/Gli signaling in the tumor microenvironment is unknown. Here, we analyzed surgically resected oral malignant melanoma specimens and observed that Sonic hedgehog, Gli1, and Gli2 were highly expressed in tumor cells, vasculatures and osteoclasts. We established a tumor bone destruction mouse model by inoculating B16 cells into the bone marrow space of the right tibial metaphysis of 5-week-old female C57BL mice. An intraperitoneal administration of GANT61 (40 mg/kg), a small molecule inhibitor of Gli1 and Gli2, caused bone destruction of cortical bone, TRAP-positive osteoclasts, and a significant suppression of endomucin-positive tumor vessels. resulted in significant inhibition of cortical bone destruction, TRAP-positive osteoclasts within the cortical bone, and endomucin-positive tumor vessels. A gene set enrichment analysis suggested that genes involved in the apoptosis, angiogenesis, and PD-L1 expression pathway in cancer were significantly altered by GANT61 treatment. A flow cytometry analysis revealed that PD-L1 expression was significantly decreased in cells in which late apoptosis was induced by GANT61 treatment. These results suggest that Hh signaling plays an important role in the melanoma bone microenvironment through normalization and remodeling of the tumor bone microenvironment. These results suggest that molecular targeting of Gli1 and Gli2 may release immunosuppression of the tumor bone microenvironment through normalization of abnormal angiogenesis and bone remodeling in advanced melanoma with jaw bone invasion.
Comment #2: Figure 1 font is different than the font mentioned in the caption. Also, the diagram's quality is not appropriate. Kindly enhance the quality of the diagrams.
Response #2: According to the reviewer’s comments, we have changed the font and increased image of the resolution.
Comment # 3: Figure 2 font is different than the font mentioned in the caption. Also, the diagram's quality is not appropriate. Kindly enhance the quality of the diagrams.
Response #3: According to the reviewer’s comments, we have changed the font and increased image of the resolution.
Comment#4: I would also suggest the author write a small conclusion deciphering the significance of their research in the field of medicine and health.
Response #4: As the reviewer pointed out, we have changed the Discussion section.
Ten years after the approval of immune checkpoint inhibitors for advanced melanoma, it is time to reflect on the lessons learned about immune system regulation in cancer treatment and consider new approaches to therapy[63]. One of the current challenges in melanoma antitumor immunotherapy is immunosuppression by the tumor microenvironment[63,64], and it is thus critical to promote remodeling of the tumor bone microenvironment, normalization of abnormal angiogenesis, and immune cell infiltration in cancer in order to enhance the antitumor effects on advanced melanoma with jaw bone invasion (Figure 6).
The reference were newly added.
- Huang, A.C.; Zappasodi, R. A Decade of Checkpoint Blockade Immunotherapy in Melanoma: Understanding the Molecular Basis for Immune Sensitivity and Resistance. Nat Immunol 2022, 23, 660–670.
Reviewer 2 Report
The research article “The Role of Hedgehog Signaling in the Melanoma Tumor Bone Microenvironment” by Shamsoon and group investigates mechanism associated with melanoma development and progression. The authors have explored tumor bone microenvironment and have studied therapeutic potential of Hedgehog (Hh) signaling as a novel target.
The data originated from experiments carried out on surgically resected oral malignant melanoma specimens and mouse model mimicking a tumor bone destruction looks satisfying. Inhibition by small molecule inhibitor GANT61 of Gli1 and Gli2 appears of high potential as it displays significant effect on bone destruction, and endomucin-positive tumor vasculature.
Finding from the research manuscript highlights the critical role of Hh signaling in the melanoma bone microenvironment via inducing a normalization and remodeling effect in the tumor bone microenvironment.
The figures are satisfactory, and the study is ethically approved. The manuscript can be accepted in its current form.
Author Response
Dear Reviewer 2
We greatly appreciate the reviewer’s insightful comments regarding our manuscript and are thankful for your considering our paper acceptable in accordance with the reviewers' comments.
Reviewer 3 Report
Major issues:
1- The Spearman's correlation shown in Fig. 1C indicates an n=40, but the number of values plotted (red dots) are just over 10. Can you please explain or re-do the plot to show all the values?
2- Why is the expression of SHH, GLI1, and GLI2 in skin melanomas determined in FIg. 1, when the study is about oral mucosal melanoma and its bone infiltration? It is essential to perform the IHCs in tumour sections of mucosal melanoma instead, and also of B16 cells that are used in the rest of the study. Do they express SHH, GLI1 and GLI2?
3- Results related to Fig. 3 show that GANT61 reduces to a similar level the expression of SHH, GLI1 , GLI2 and PD-L1. This is very unusual, given that as a GLI1/2 inhibitor, it shouldn't affect SHH expression, which is independent of GLI transcriptional activity, and could be a direct consequence of the cytotoxic effect of GANT61 on B16 cells (fewer cells surviving = fewer staining of any marker). These data should be analysed by double staining of Mel-A and the Hh markers to determine if there is a reduction of the markers or if it reflects the surviving population of melanoma cells.
4- Treatment of B16 cells with GANT61 in vitro should include analysis of the markers studied in vivo in the bone microenvironmnet. In particular, the authors should include RT-PCR data (and microarray, if available) of levels of SHH, GLI1, GLI2 and PD-L1 in DMSO vs GANT61 treated cells at 20 uM. Without this data, it is impossible to determine if the effect of GANT61 on these cells is Hh-pathway specific.
5- The analysis and interpretation of the effect of GANT61 on apoptosis and its correlation with PD-L1 expression do not make much sense. First, any role of PD-L1 on apoptosis will require the presence of cytotoxic T cells in the system. Second, how can the authors conclude that GANT 61 increases late apoptosis and not early apoptosis? Cells must go through early apoptosis before they reach advanced stages... Third, the correlation of the two stages with PD-L1 is nonsensical. I suggest to remove all mention of PD-L1 in this figure, and possibly add a cell cycle profile to understand better the mechanism behind the cytotoxicity of GANT61.
6- The discussion about the GSEA term GPCRs is too speculative. Unless there is a table showing which GPCRs are altered by GANT61 and in which direction (up or down), a proper explanation cannot be given.
7- it is erroneous to affirm in the discussion that the Hh pathway regulates PD-L1 expression in B16 cells. At least an RT-PCR result showing significant changes in PD-L1 expression in cells treated with GANT61 should be presented.
Minor issues:
1- The sentence in introduction "The expression and the mechanisms of GANT61 in the melanoma bone microenvironment" is unclear. Please re-phrase.
2- Line 249:"These results suggest that GANT61 suppresses tumor cell numbers by inhibiting Hh signaling in the bone microenvironment. We thus next performed an immunohistochemical analysis of the effects of GANT61 on the expressions of SHH, Gli1, and Gli2 in the tumour bone microenvironment" needs re-writing. The results so far suggest that GANT61 suppresses tumour cell numbers in the bone microenvironment, which then lead to investigate if it was through inhibition of canonical Hh signalling...
3- Table S4 is duplicated
Author Response
Dear Reviewer 3
We greatly appreciate the reviewer’s insightful comments regarding our manuscript and are thankful for your considering our paper acceptable in accordance with the reviewers' comments. We have answered all of the reviewers’ comments below and modified the text and added reference accordingly.
Major issues:
Comment #1: The Spearman's correlation shown in Fig. 1C indicates an n=40, but the number of values plotted (red dots) are just over 10. Can you please explain or re-do the plot to show all the values?
Response #1: We have added the text in the 4.1 Tissue array analysis in the Materials and Methods and Figure legend sections.
There were 3 samples with tissue detached, analyzed as n=37
Staining intensity (SI, A) was evaluated visually: negative (0), weak (1), moderate (2), and strong (3). SI3 (SHH and Gli1, n=15), SI3 (SHH and Gli2, n=12), SI2 (SHH and Gli1, n=7) and SI2 (SHH and Gli2, n=4).
Comment #2: Why is the expression of SHH, GLI1, and GLI2 in skin melanomas determined in FIg. 1, when the study is about oral mucosal melanoma and its bone infiltration? It is essential to perform the IHCs in tumour sections of mucosal melanoma instead, and also of B16 cells that are used in the rest of the study. Do they express SHH, GLI1 and GLI2?
Response #1: We have added the text in the Results sections of 2.1 Immunohistochemical expressions of SHH, Gli1, and Gli2 on the human melanoma samples.
Before analyzing the mouse model of tumor bone destruction using skin-derived B16 mouse melanoma cells, a screening analysis of Sonic Hedgehog (SHH) signaling was performed using skin melanoma resection specimens without bone involvement (n=37). To determine whether soft tissuemelanoma expresses Sonic Hedgehog (SHH) and its signals Gli1 and Gli2, we performed an immunohistochemical analysis of excised human melanoma specimens and normal skin tissue, and we observed the expressions of SHH, Gli1 and Gli2 not only on the melanoma cells but also in tumor vascular endothelial cells in the stroma (Figure 1A).
Comment #3: Results related to Fig. 3 show that GANT61 reduces to a similar level the expression of SHH, GLI1, GLI2 and PD-L1. This is very unusual, given that as a GLI1/2 inhibitor, it shouldn't affect SHH expression, which is independent of GLI transcriptional activity, and could be a direct consequence of the cytotoxic effect of GANT61 on B16 cells (fewer cells surviving = fewer staining of any marker). These data should be analysed by double staining of Mel-A and the Hh markers to determine if there is a reduction of the markers or if it reflects the surviving population of melanoma cells.
Response #3: As the reviewer pointed out, we have used a labeling index (LI) to determine the percentage of SHH, Gli1 and Gli2 positive cells in Melan-A-positive tumor cells/mm2 at 400x magnification, and was regarded as the LI as described in the Materials and Methods.
For the determination of the LI, the percentage of positive cells in Melan-A-positive tumor cells/mm2 was observed at ×400 magnification and was regarded as the LI.
Figure 3B-F
Vertical axes of graphs in Figure3B-F were modified to PCNA LI (%), SHH LI (%), Gli1 LI (%), Gli2 LI (%) and PD-L1 LI (%)。
The description of Result has also been modified as follows.
These results suggest that GANT61 suppresses tumor cell numbers by inhibiting Hh signaling in the bone microenvironment. We thus next performed an immunohistochemical analysis of the effects of GANT61 on the expressions of SHH, Gli1, and Gli2 in the tumor bone microenvironment, and SHH-expressing cells in tumor tissue present in the bone marrow were observed when B16 cells had been inoculated into the tibial metaphysis (Figure 3C). The number of SHH-expressing cells in tumor tissue was decreased in the GANT61-treated group (Figure 3C), and thepercentage of SHH-expressing cells in tumor tissue per mm2 -positive cells in Melan-A-positive tumor cells was significantly decreased in the GANT61-treated group (LI: 45% ± 5.08%) compared to the control group (LI: 65% ± 2.63%) (Figure 3C, p<0.05). SHH staining of the contralateral tibial metaphysis without B16 cells was performed for comparison, and very few SHH-positive cells were detected in the bone marrow cells (Figure 3C).
Regarding the expressions of Gli1 and Gli2, we observed Gli1 and Gli2 expression in the tumor tissue in the bone marrow in addition to the SHH expression. In contrast, Gli1- (Figure 3D) and Gli2- (Figure 3E) expressing cells in the tumor tissue were decreased in the GANT61-treated group. The the percentage of Gli1- and Gli2-expressing cells per mm2 -positive cells in Melan-A-positive tumor cells was significantly decreased in the GANT61-treated group compared to the control group (Gli1: Figure 3D, p<0.05, Gli2: Figure 3E, p<0.05). Similarly, Gli1 and Gli2 staining of the contralateral tibial metaphyses without B16-cell inoculation for comparison showed hardly any Gli1- or Gli2-positive cells in the bone marrow cells (Figure 3D,E). An immunohistochemical analysis of the effect of GANT61 on PD-L1 expression in the tumor bone microenvironment was thus performed, and it revealed that when B16 cells were inoculated into the tibial metaphysis of mice, PD-L1-expressing cells in tumor tissue present in the bone marrow were recognized (Figure 3F). The number of PD-L1-expressing cells in tumor tissue was decreased in the GANT61-treated group (Figure 3F), and the percentage of PD-L1-expressing cells in tumor tissue per unit area -positive cells in Melan-A-positive tumor cells was significantly decreased in the GANT61-treated group as indicated by the LI (34% ± 2.23%) compared to the control group's LI (72% ± 2.42%) (Figure 3F, p<0.05). PD-L1 staining of the contralateral tibial metaphyses without B16 cells was performed for comparison, and very few PD-L1 positive cells were detected in the bone marrow cells (Figure 3F). These results suggest that GANT61 is involved in the reduction of SHH, Gli1, Gli2, and PD-L1 expression in surviving melanoma cells in the bone microenvironment.
The description of Figure legends has also been modified as follows.
Figure 3. Immunohistochemical analysis of Melan-A, proliferating cell nuclear antigen (PCNA), SHH, Gli1, Gli2 and PD-L1 in bone from mice bearing B16 melanoma cells. (A, B) Melan-A (A) and PCNA (B) expression in the tibial metaphysis with (right) or without (left) B16-cell inoculation treated with DMSO (upper) or GANT61 (lower). Each image is a magnification of the typical cortical bone area (lower parts) of the corresponding upper parts images in panel A and B. Scale bar: 500 µm (upper parts). Scale bar: 50 µm (lower parts). The number of Melan-A-positive tumor cells per mm2 (A, graph, n=5/group). The percentage of PCNA-positive cells/Melan-A-positive tumor cells/mm2 (n=5). (C-F) The expression of SHH (C), Gli1(D), Gli2 (E) and PD-L1 (F) at the typical cortical bone are in tibial metaphyses with (right) or without (left) B16-cell inoculation treated with DMSO (upper) or GANT61 (lower). Scale bar: 50 µm. The percentages of SHH (C), Gli1(D), Gli2 (E) and PD-L1 (F)-positive cells/Melan-A-positive tumor cells/mm2 (n=5). Data are mean ± SD. Arrowhead: bone rupture. Arrow: bone resorption. CB: cortical bone. LI: Labeling index. *p<0.05 between the indicated groups.
Comment #4: Treatment of B16 cells with GANT61 in vitro should include analysis of the markers studied in vivo in the bone microenvironmnet. In particular, the authors should include RT-PCR data (and microarray, if available) of levels of SHH, GLI1, GLI2 and PD-L1 in DMSO vs GANT61 treated cells at 20 uM. Without this data, it is impossible to determine if the effect of GANT61 on these cells is Hh-pathway specific.
Response #4: As the reviewer pointed out, we have added our results information and the comments with new references in the Discussion section. We have also changed the Figure 6..
However, the direct effect of GANT61 on individual SHH, Gli1 and Gli2 in melanoma cells were not evident with no significant change unlike in vivo data. Tumor cells need large amounts of nutrients and growth factors that are supplied and distributed to the tumor tissue by the aberrant tumor vasculature[57]. It was suggested that growth factors including SHH produced by tumor cells paracrinely induce Gli1 and Gli2 expression in tumor vascular endothelial cells, and various angiocline factors produced by tumor vascular endothelial cells induce SHH and Gli1 in tumor cells (Figure 6). In the melanoma bone microenvironment, mutual crosstalk between tumor cells, tumor vascular endothelial cells and osteoclasts is implicated in the expression of Hh signaling in tumor cells[58,59].
New references were added.
- Lidonnici, J.; Santoro, M.M.; Oberkersch, R.E. Cancer-Induced Metabolic Rewiring of Tumor Endothelial Cells. Cancers (Basel) 2022, 14.
- Zarrer, J.; Haider, M.T.; Smit, D.J.; Taipaleenmäki, H. Pathological Crosstalk between Metastatic Breast Cancer Cells and the Bone Microenvironment. Biomolecules 2020, 10.
- Alsina‐sanchis, E.; Mülfarth, R.; Fischer, A. Control of Tumor Progression by Angiocrine Factors. Cancers (Basel) 2021, 13.
Comment #5: The analysis and interpretation of the effect of GANT61 on apoptosis and its correlation with PD-L1 expression do not make much sense. First, any role of PD-L1 on apoptosis will require the presence of cytotoxic T cells in the system. Second, how can the authors conclude that GANT 61 increases late apoptosis and not early apoptosis? Cells must go through early apoptosis before they reach advanced stages... Third, the correlation of the two stages with PD-L1 is nonsensical. I suggest to remove all mention of PD-L1 in this figure, and possibly add a cell cycle profile to understand better the mechanism behind the cytotoxicity of GANT61.
Response #5: As suggested by reviewer, we stated about it in Flow cytometry of Materials and Methods, and Fig. 5 of Figure legends. As reported by Alexandraki, A, et al. [70], the relationship between apoptosis and PD-L1 has been demonstrated separately from the presence of cytotoxic T cells. The methods for detecting early and late apoptosis have been reported by Vermes, I, et al. [69]. We described that the relationship between early apoptosis, late apoptosis, and PD-L1 by GANT 61 was hierarchically linked as P4 to P5 to P6 of gating in Fig. 5.
Result section
- Proportion of apoptotic cells
The GSEA analysis revealed the possibility that GANT61 regulates apoptosis and PD-L1 expression signaling in B16 cells, and we thus next examined the association between apoptosis and PD-L1-expressing cells by flow cytometry. The percentages of cells stained with PI-negative andannexin V-positive were 0.5% in the control group and 0.8% in the GANT61-treated group, indicating that GANT61 treatment increased early apoptosis (Figure 5A, p=0.058). The percentage of PD-L1-positive cells among those cells stained with PI-negative and annexin V-positive was 0.0% in the GANT61-treated group versus 6.7% in the control group, revealing a trend toward decreased PD-L1 expression in the cells in which early apoptosis was induced by GANT61 treatment (Figure 5A, p=0.134). In contrast, the percentages of PI-positive and annexin V-positive cells were 11.4% in the control group and 14.2% in the GANT61-treated group, indicating that GANT61 significantly promoted late apoptosis (Figure 5B, p=0.000). The percentages of PD-L1-positive cells among those cells stained with PI-positive and annexin V-positive were 92.8% in the control group and 88.7% in the GANT61-treated group, indicating that (i) GANT61-treated cells underwent late apoptosis, and (ii) the PD-L1 expression was significantly decreased in cells in which late apoptosis was induced by GANT61 treatment (Figure 5B, p=0.000).
Figure 5. Flow cytometry results of the annexin-V and propidium iodide (PI) staining of apoptotic cells following the GANT61 treatment of B16 cells. Each population was hierarchically linked as P4 to P5 to P6 of gating. The early apoptosis was defined as PI-negative and annexin V-positive, and the late apoptosis was defined as PI-positive and annexin V-positive. Furthermore, PD-L1-positive cells were counted among those cells by gating. (A) Early apoptosis: No significant difference in the number of cells stained as PI- negative and annexin V-positive was observed between the control and GANT61 groups (χ2-test, p=0.058, n=4). No significant difference in the number of cells stained with PD-L1-positive among those earlyapoptotic cells was detected between the control and GANT61 groups (χ2-test, p=0.134, n=4). (B) Late apoptosis: A significant increase in the number of cells stained PI-positive and annexin V-positive was observed in the GANT61 group compared to the control group (χ2-test, **p=0.000, n=4). A significant decrease in the number of cells stained with PD-L1-positive among those late apoptotic cells was detected in the GANT61-treated group compared to the control group (χ2-test, **p=0.000, n=4).
The references were newly added.
- Vermes, I.; I Haanen, C.; Steffens-Nakken, H.; Reutelingsperger, C. A novel assay for apoptosis. Flow cytometric detection of phosphatidylserine expression on early apoptotic cells using fluorescein labelled Annexin V. J Immunol Methods 1995, 184, 39-51, doi:10.1016/0022-1759(95)00072-i.
- Alexandraki, A.; Strati, K. Decitabine Treatment Induces a Viral Mimicry Response in Cervical Cancer Cells and Further Sensitizes Cells to Chemotherapy. Int J Mol Sci 2022, 23, doi:10.3390/ijms232214042.
Comment #6: The discussion about the GSEA term GPCRs is too speculative. Unless there is a table showing which GPCRs are altered by GANT61 and in which direction (up or down), a proper explanation cannot be given.
Response #6: As the reviewer pointed out, we have changed the Discussion section.
The present study's GO functional category analysis revealed that the GRCR GPCR signaling pathways was were significantly altered by GANT61 treatment (Table S2, p=0.000051). Melanomas in the bone microenvironment may thus "hijack" the normal physiological functions of GPCRs via Hhsignaling for malignant behavior. Melanomas in the bone microenvironment may thus "hijack" the normal physiological functions of GPCRs via Hhsignaling for malignant behavior. The direct effect of GANT61 on individual GPCRs in melanoma cells were not evident using Z-scores and ratios with no significant changes. However, to reveal a comprehensive paradigm of GANT61 function, gene ontology (GO) functional category analysis revealed that the GRCR signaling pathways were significantly altered by GANT61 treatment (Table 2, p=0.000051).
Comment #7: it is erroneous to affirm in the discussion that the Hh pathway regulates PD-L1 expression in B16 cells. At least an RT-PCR result showing significant changes in PD-L1 expression in cells treated with GANT61 should be presented.
Response #7: As pointed out by the reviewer, we have added the text and reference in Discussion section.
Our GSEA analysis data indicate that Hh signaling plays a role in mediating PD-L1 expression in melanoma cells. The increased expression of PD-L1 on the tumor cells evades immune surveillance by upregulating the surface expression of PD-L1, which interacts with PD-1 on T cells to trigger immune checkpoint responses[45,46]. Anti-PD-1 antibodies have also shown remarkable promise in the treatment of metastatic melanoma[47]; however, the response rate for patients is still low[48,49]. The present study's GSEA analysis data indicate that Hh signaling plays a role in mediating the expression of PD-L1 in melanoma cells (Figure 4D). In support of these findings, a study using pancreatic ductal adenocarcinoma cell and gastric cancer cells revealed cell lines that showed Hh signaling-induced PD-L1 expression[50], increased Hh activity correlated with multiple immunosuppressive characteristics in the tumor microenvironment of diverse cancers[51]. We determined the direct effect of GANT61 on PD-L1 gene expression in melanoma cells using p-values or log2 fold-change from differential expression results of GSEA analysis to identify whether gene sets work together in a coordinated manner. Although no significant change of PD-L1 expression by GANT61 was observed in Z-scores and ratios, GSEA analysis revealed that many genes with low fold-change by GANT61 was found to have a significant effect in concert with PD-L1 expression and PD-1 checkpoint pathway in melanoma.
[51] reference was added.
- Jiang, J.; Ding, Y.; Chen, Y.; Lu, J.; Chen, Y.; Wu, G.; Xu, N.; Wang, H.; Teng, L. Pan-Cancer Analyses Reveal That Increased Hedgehog Activity Correlates with Tumor Immunosuppression and Resistance to Immune Checkpoint Inhibitors. Cancer Med 2022, 11, 847–863, doi:10.1002/cam4.4456.
Minor issues:
Comment #1: The sentence in introduction "The expression and the mechanisms of GANT61 in the melanoma bone microenvironment" is unclear. Please re-phrase.
Response #1: Thank you for pointing that out. We have deleted the sentence at the Introduction section.
The expression and the mechanisms of GANT61 in the melanoma bone microenvironment were also evaluated to determine whether the microenvironment could be a therapeutic potential target following bone invasion of melanoma.
Comment #2: Line 249:"These results suggest that GANT61 suppresses tumor cell numbers by inhibiting Hh signaling in the bone microenvironment. We thus next performed an immunohistochemical analysis of the effects of GANT61 on the expressions of SHH, Gli1, and Gli2 in the tumour bone microenvironment" needs re-writing. The results so far suggest that GANT61 suppresses tumour cell numbers in the bone microenvironment, which then lead to investigate if it was through inhibition of canonical Hh signalling...
Response #2: Thank you for correcting the sentence. We have deleted the sentence and added the text at the Results section.
These results suggest that GANT61 suppresses tumor cell numbers by inhibiting Hh signaling in the bone microenvironment. We thus next performed an immunohistochemical analysis of the effects of GANT61 on the expressions of SHH, Gli1, and Gli2 in the tumor bone microenvironment, The results so far suggest that GANT61 suppresses tumor cell numbers in the bone microenvironment, which then lead to investigate if it was through inhibition of canonical Hh signaling. and SHH-expressing cells in tumor tissue present in the bone marrow were observed when B16 cells had been inoculated into the tibial metaphysis (Figure 3C).
Comment #3: Table S4 is duplicated
Response #3: Thank you for pointing that out. We have deleted the Table S4.
Round 2
Reviewer 1 Report
The authors have incorporated all the required suggestions and improved the quality of the manuscript in an efficient manner. Hence, the manuscript can be accepted for publication.